# *OPA1* Dominant Optic Atrophy: Diagnostic Approach in the Pediatric Population

Natalia Arruti [1,2,†], Patricia Rodríguez-Solana [3,†], María Nieves-Moreno [1,2], Marta Guerrero-Carretero [1], Ángela del Pozo [4,5], Victoria E. F. Montaño [3,4], Fernando Santos-Simarro [4,6,7], Emi Rikeros-Orozco [6], Luna Delgado-Mora [6], Elena Vallespín [2,3,4,*] and Susana Noval [1,2]

1 Department of Pediatric Ophthalmology, IdiPaz, Hospital Universitario La Paz, 28046 Madrid, Spain
2 European Reference Network on Eye Diseases (ERN-EYE), La Paz University Hospital, 28046 Madrid, Spain
3 Molecular Ophthalmology Section, Medical and Molecular Genetics Institute (INGEMM) IdiPaz, Hospital Universitario La Paz, 28046 Madrid, Spain
4 Biomedical Research Center in the Rare Diseases Network (CIBERER), Carlos II Health Institute (ISCIII), 28029 Madrid, Spain
5 Clinical Bioinformatics Section, Medical and Molecular Genetics Institute (INGEMM) IdiPaz, CIBERER, Hospital Universitario La Paz, 28046 Madrid, Spain
6 Clinical Genetics Section, Medical and Molecular Genetics Institute (INGEMM) IdiPaz, CIBERER, Hospital Universitario La Paz, 28046 Madrid, Spain
7 Molecular Diagnostics and Clinical Genetics Unit, Health Research Institute of the Balearic Islands (IdISBa), Hospital Universitari Son Espases, 07120 Palma, Spain
* Correspondence: elena.vallespin@salud.madrid.org
† These authors contributed equally to this work.

**Abstract:** A clinical and genetic study was conducted with pediatric patients and their relatives with optic atrophy 1 (*OPA1*) mutations to establish whether there is a genotype–phenotype correlation among the variants detected within and between families. Eleven children with a confirmed *OPA1* mutation were identified during the study period. The main initial complaint was reduced visual acuity (VA), present in eight patients of the cohort. Eight of eleven patients had a positive family history of optic atrophy. The mean visual acuity at the start of the study was 0.40 and 0.44 LogMAR in the right and left eye, respectively. At the end of the study, the mean visual acuity was unchanged. Optical coherence tomography during the first visit showed a mean retinal nerve fiber layer thickness of 81.6 microns and 80.5 microns in the right and left eye, respectively; a mean ganglion cell layer of 52.5 and 52.4 microns, respectively, and a mean central macular thickness of 229.5 and 233.5 microns, respectively. The most common visual field defect was a centrocecal scotoma, and nine out of eleven patients showed bilateral temporal disc pallor at baseline. Sequencing of *OPA1* showed seven different mutations in the eleven patients, one of which, NM_130837.3: c.1406_1407del (p.Thr469LysfsTer16), has not been previously reported. Early diagnosis of dominant optic atrophy is crucial, both for avoiding unnecessary consultations and/or treatments and for appropriate genetic counseling.

**Keywords:** dominant optic atrophy; Kjer type; *OPA1*; mitochondrial neuropathies

## 1. Introduction

Dominant optic atrophy (DOA) is a mitochondrial optic neuropathy characterized by optic nerve fiber degeneration. DOA is a genetic condition also known as autosomal dominant optic atrophy (ADOA) due to its mode of inheritance, in contrast to Leber hereditary optic neuropathy (LHON), which is caused by pathogenic variants in the mitochondrial deoxyribonucleic acid (DNA). DOA is the most commonly diagnosed inherited optic neuropathy, with a prevalence varying from 1/12,000 to 1/50,000, depending on the series [1], compared with LHON whose prevalence in a recent meta-analysis was 1/40,000 in Europe [2]. DOA usually begins in the first or second decade of life with progressive bilateral visual loss, whereas LHON usually begins one decade later with acute painless visual loss [1].

*OPA1* mutations account for up to 60–80% of patients with DOA [3]. The nuclear gene *OPA1* is located on chromosome 3q29 and encodes a mitochondrial protein similar to dynamin-related GTPases with a N-terminal mitochondrial targeting sequence, which directs the protein to the inner membrane or intermembrane space [4] (RefSeq: NM_130837.3, *OPA1* transcript variant 8).

In humans, the *OPA1* gene (NG_011605.1) is built from 31 exons, three of which (4, 4b and 5b) are alternatively spliced, leading to eight variants (isoforms 1–8) with tissue-specific patterns of expression. After translation, Opa1 precursors are imported into the mitochondria, where mitochondrial processing peptidase cleaves the N-terminal targeting sequence giving rise to the long isoforms (L-Opa1), which remain anchored to the inner mitochondrial membrane on an N-terminal transmembrane segment [3]. Approximately half of L-Opa1 are cleaved at two intramitochondrial proteolytic cleavage sites by either of the inner membrane-anchored proteases OMA1 or YMEL1 to produce different non-membrane-anchored short forms (S-Opa1). OMA1 cleaves at the S1 site encoded by exon 5, and YME1L cleaves at the S2 site encoded by the alternative exon 5b [5].

OPA1 plays essential roles in mitochondria, including remodeling cristae morphology, inner mitochondrial membrane fusion, efficiency of oxidative phosphorylation, and mitochondrial genome maintenance. Under normal conditions, there is a stoichiometric balance between L-Opa1 and S-Opa1, and each function is performed by specific isoforms [6].

Typically, hereditary neuropathies present with reduced visual acuity (VA), cecocentral scotomas in the visual field, and temporal disc pallor [7]. Significant decreases in both the ganglion cell layer and retinal nerve fiber layer thickness are detected in patients with DOA or LHON with differing patterns [8]. Tritanopia, described as blue-yellow color blindness, has classically been a key in differentiating OPA1 from other neuropathies in which the red–green axis is affected [9]. Certain patients with DOA present with normal VA and normal fundus, due to its incomplete penetrance, explaining the variety of phenotypes found in these patients [10].

DOA can be isolated or syndromic, frequently called DOA plus, and is associated with extraocular symptoms, typically neuromuscular, such as sensorineural hearing loss, myopathy, and peripheral neuropathy, commonly found in mitochondriopathies [11].

The therapeutic approach is limited in inherited optic neuropathies. Idebenone is a potent antioxidant that interacts with the mitochondrial electron transport chain, thereby facilitating mitochondrial electron flux overcoming complex I. Idebenone was approved by the European Medicines Agency in 2015 for treating visual impairment in adolescent and adult patients with LHON. Due to pathophysiological similarities between LHON and DOA, visual stabilization and/or improvement in patients with DOA treated with off-label idebenone have since been reported [12,13].

The aim of this study was to establish whether there is a genotype–phenotype correlation in patients with DOA according to the variants detected within and between families.

## 2. Results

### 2.1. Ophthalmological Examination

Eleven patients were included in this study, all of whom were Caucasian except for two, who had Asian ethnicity. There was a male predominance, with only four female patients. The mean age at diagnosis was 9 years (range, 2–15 years), and there were 4 pairs of siblings. The most common reason for consultation was reduced VA, present in eight patients. One patient was asymptomatic, one had esotropia, and another patient was examined after his brother was diagnosed with DOA. Eight of the patients had a positive family history of DOA. Our cohort's demographic data are shown in Table 1.

Mean baseline VA was 0.40 and 0.44 LogMAR in the right and left eye, respectively. The final mean VA was 0.40 and 0.44 LogMAR, respectively.

Colour vision was assessed in ten patients, and only one showed tritanopia.

**Table 1.** Demographic characteristics of our cohort.

| Family ID | Individual ID | Sex | Race | Age at Diagnosis, Years | Present Complaint | Family History | Color Vision |
|-----------|---------------|-----|------|-------------------------|-------------------|----------------|--------------|
| OFT-00133 | 2620605 | M | Caucasian | 6 | Reduced VA | Yes | Normal |
| OFT-00536 | 2624905 | M | Asian | 9 | Esotropia | Yes | Normal |
| OFT-00536 | 2235284 | M | Asian | 14 | Reduced VA | Yes | Normal |
| OFT-00560 | 3338850 | F | Caucasian | 9 | Reduced VA | No | Normal |
| OFT-00560 | 3338857 | F | Caucasian | 15 | Asymptomatic | Yes | Normal |
| OFT-00615 | 3370622 | M | Caucasian | 8 | Reduced VA | Yes | Normal |
| OFT-00615 | 3395589 | M | Caucasian | 15 | Reduced VA | Yes | Normal |
| OFT-00641 | 3346308 | M | Caucasian | 4 | Reduced VA | No | Tritanopia |
| OFT-00641 | 3366708 | M | Caucasian | 2 | Brother DOA | Yes | No cooperationn |
| OFT-00677 | 3408438 | F | Caucasian | 14 | Reduced VA | No | Normal |
| OFT-00786 | 2854654 | F | Caucasian | 7 | Reduced VA | Yes | Normal |

Abbreviations: M, Male; F, Female; VA, visual acuity; DOA, dominant optic atrophy.

Intraocular pressure was measured in all our patients and was within normal limits. Mean values were 16.5 and 17.0 mmHg in the right and left eye, respectively.

Confrontation visual field testing showed no gross defect in any patient. The Humphrey visual field test was performed in nine patients, and six showed a cecocentral scotoma.

Fundoscopy showed bilateral temporal disc pallor in nine patients. The remaining two patients had normal discs. These findings were confirmed with retinography.

None of the patients had systemic symptoms during the study period.

*2.2. Optic Coherence Tomography*

The mean retinal nerve fiber layer (RNFL) thickness at baseline was 81.6 and 80.5 µm in the right and left eye, respectively. The mean temporal sector thickness at the first visit was 45.4 and 49.4 µm, respectively. The mean RNFL thickness at the last visit was 76.4 and 75 µm, respectively, whereas the mean temporal sector thickness at the last visit was 45.5 and 43.2 µm, respectively.

The mean ganglion cell layer (GCL) thickness at baseline was 52.5 and 52.4 µm, respectively. At the last visit, the mean GCL thickness was 52.1 and 52.5 µm, respectively.

The mean central macular thickness (CMT) was 229.5 and 233.5 µm, respectively, at the first visit and 236.5 and 232.7 µm, respectively, at the last visit. There was a strong positive correlation ($R^2 = 0.712$) between GCL thickness and VA in our cohort, whereas there was no correlation between RNFL thickness and VA (Figure 1).

The RNFL and GCL thickness values for the patients with DOA were compared with the normative database of healthy Spanish children published by Barrio-Barrio and colleagues [14]. The mean RNFL thickness, temporal RNFL thickness, and mean GCL thickness were smaller in the patients with DOA ($n = 11$) than in the healthy children ($p < 0.001$). Analyzing the group of children younger than 11 years ($n = 5$), the temporal RNFL thickness and mean GCL thickness were smaller in the younger patients with DOA than in the healthy children. There were no significant differences between the patients with DOA younger than 11 years and the healthy children in terms of mean RNFL thickness ($p = 0.38$).

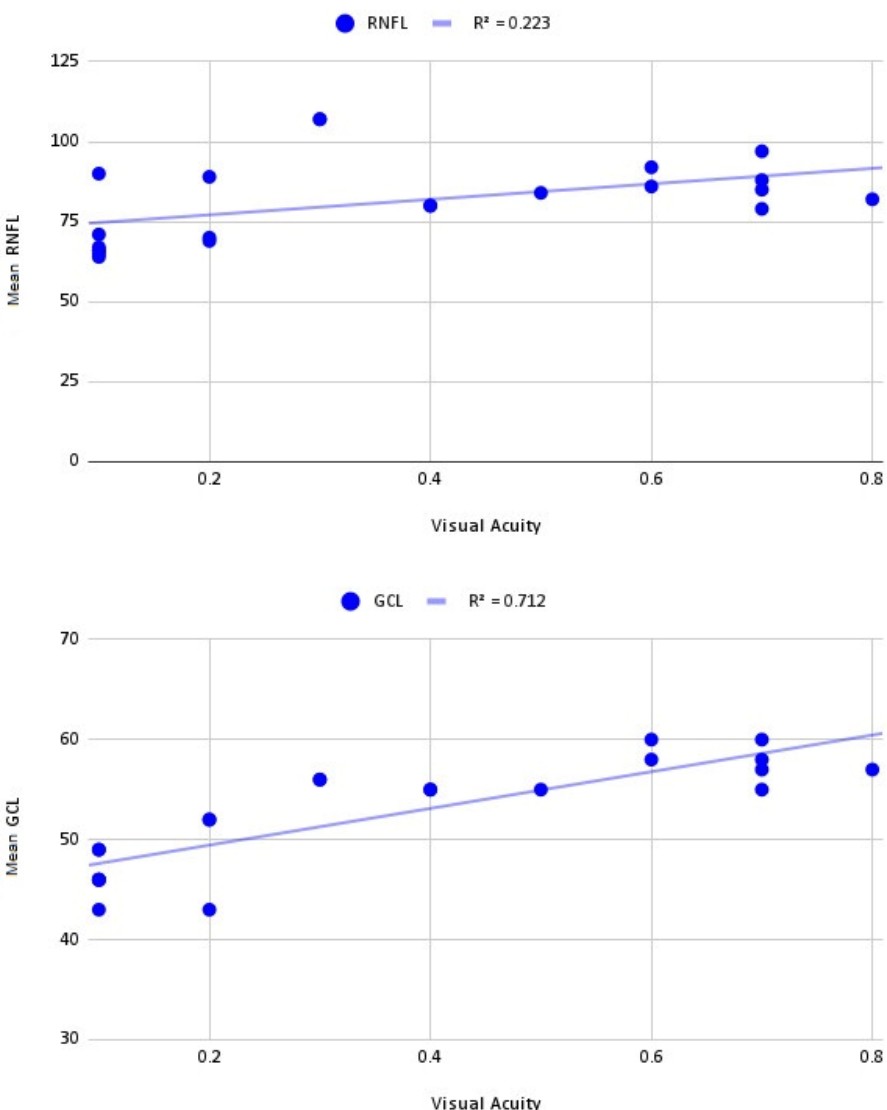

**Figure 1.** Correlation between mean retinal nerve fiber layer—ganglion cell layer thickness and visual acuity.

*2.3. Molecular Genetics*

After performing an NGS analysis with a specifically designed panel, we found seven variants in seven probands analyzed in *OPA1* (Table 2 and Figure 2). The analyzed families are listed below.

2.3.1. OFT-00133

This patient was one of the two patients with the most severe phenotype.

He is heterozygous for the mutation c.267G>A(p.Trp89Ter) (NC_000003.11:g.193332746), which generates a premature STOP codon that can cause loss of normal protein function by truncation or nonsense-mediated mRNA decay (NMD) resulting in an absent or altered protein product. NMD is a widespread cellular process that screens nascent mRNA transcripts and destroys those carrying premature termination codons before they are translated into truncated and potentially harmful proteins [15]. In 2008, Schimpf et al. [16] showed that most *OPA1* nonsense mutations underwent NMD. Specifically, this variant has been described by one submitter in ClinVar, accession: VCV000982032.1 and it has been reported by Le Roux et al. [17]. Furthermore, it is a null variant in a gene where loss of function (LoF) is a known mechanism of disease (pLi = 0.99). In fact, a stop-gained mutation that affects the above amino acid has been reported once in gnomAD (NC_000003.11g.3-193332743) as pathogenic. In addition,

patients' mutation allele frequency in gnomAD population databases is extremely low (0.0%). Therefore, the c.267G>A(p.Trp89Ter) (NC_000003.11:g.193332746) variant has been classified as likely pathogenic.

### 2.3.2. OFT-00536

The youngest brother presented with a partially accommodative esotropia. At the first visit, the patient showed no signs of other ocular conditions. As his VA started to decrease and his visual field showed a central scotoma, however, further investigations were performed. The eldest brother did not complain of a visual deficit, but the segregation study confirmed the segregation of the same variant found in his affected brother.

The patient and his brother are heterozygous for the variant c.2873_2876del (p.Val958GlyfsTer3) (NC_000003.11:g.193384956). The deletion results in a change in the reading frame generating a premature STOP codon three positions forward. This variant has been reported as pathogenic in ClinVar by several groups and has been previously reported in families with optic atrophy type 1 [9,18]. The patient presented by Wang et al. [18] with the c.2873_2876del variant had a best-corrected VA of 0.4 in the right eye and 0.3 in the left eye and a pale optic disc.

Delettre et al. [9] examined the pathophysiological effect of changes in the C-terminal by studying monocytes from patients with the c.2873_2876del mutation. The authors found that the distribution of the patients' mitochondria was abnormal, altering their integrity and structure. Moreover, the loss-of-function variants in *OPA1* are known to be pathogenic. For these reasons, coupled with its extremely low population frequency in databases (gnomAD), this variant has been classified as pathogenic.

### 2.3.3. OFT-00560

This family was diagnosed as the result of the study of the youngest sister due to mild loss of VA. This is an example of incomplete penetrance, given that this mutation was inherited from their father who showed no signs of optic atrophy on funduscopy; even his GCL thickness was within normal limits.

Both sisters (3338850, 3338857) are heterozygous for the mutation c.1406_1407del (p.Thr469LysfsTer16) (NC_000003.11:g.193361341) inherited from their unaffected father. The deletion causes a frameshift starting with the codon threonine 469, changes this amino acid to a lysine residue and creates a premature stop codon at position 16 of the new reading frame, which can result in a truncated protein or the absence of protein product affecting mitochondrial function. This variant is infrequent in the population (GnomAD) and has not been previously described in the literature. On this basis, the variant has been classified as pathogenic according to the American College of Medical Genetics and Genomics (ACMG) criteria.

### 2.3.4. OFT-00615

These siblings show all the signs typically described in patients severely affected by DOA. The eldest brother is one of the two patients who showed a progressive deterioration in VA.

Both siblings (3370622, 3395589) are heterozygous for the mutation c.1041_1043del (p.Val349del) (NC_000003.11:g.193355740) inherited from their mother, who also has the variant in heterozygosity and has optic atrophy type 1. This is an in-frame variant that results in an amino acid deletion (Val) and has been previously reported in an Italian [19] and Greek [20] family, respectively. In the latter article [20], three families carrying the c.1041_1043del mutation presented with temporal optic nerve pallor and VA of 1.0/1.3, 0.1/0.2, and 0.4/0.3 (right/left eye). Various bioinformatic predictors of splicing (MaxEnt, NNSPLICE, SSF) predicted the change in the 6 bps upstream acceptor site (+197.4%), affecting a critical domain in protein function, which could ultimately trigger a deletion or truncation in the final product. Furthermore, the presence of this mutation in gnomAD

population databases is extremely low. Based on this evidence, it has been classified as likely pathogenic according to the ACMG guidelines.

### 2.3.5. OFT-00641

These siblings were the youngest patients in our case series and appeared to have severely reduced VA. However, this needs to be confirmed throughout their follow-up. Their mean RNFL thickness was normal, whereas the GCL thickness was reduced.

The nonsense variant c.2013G>A(p.Trp671Ter) (NC_000003.11:g.193372651) was present in heterozygosity in both siblings (3346308, 3366708) and in their father who also has optic atrophy type 1. The TGG codon is changed to a premature STOP codon that interrupts the reading frame. Protein truncation or nonsense-mediated mRNA decay can cause loss of normal protein function resulting in an absent or altered protein product. *OPA1* is intolerant to haploinsufficiency and LoF is a known mechanism of disease (pLi = 0.99); hence, that null mutation is expected to generate a pathogenic effect. Furthermore, it was reported in 2009 in a screening of 980 cases of suspected hereditary optic neuropathy [21]. In addition to this evidence, its extremely low frequency in gnomAD population databases classifies it as pathogenic according to the ACMG guidelines.

### 2.3.6. OFT-00677

This female patient was diagnosed at 14 years of age. Her VA was mildly reduced whereas the rest of her visual function was within normal limits. The diagnosis was suspected as she presented with bilateral temporal disc pallor and atrophy, which was confirmed on optical coherence tomography (OCT).

This patient is heterozygous for the mutation c.1099C>T(p.Arg367Ter) (NC_000003.11: g.193355804), which is in the crucial GTPase domain, and, as in the previous case, the substitution of cytosine for thymine results in the generation of a premature STOP codon. The predicted effect of this disruption would be to severely shorten the 960 amino acid wild-type protein to 311 amino acids. This variant was first reported in 2005 in two siblings with optic atrophy type 1 [22]; since then, other researchers have reported it in association with DOA [23,24]. Specifically, Pretegiani [23] presented a 10-year-old boy with a VA of 0.4/0.3 (according to the Snellen scale [right eye, left eye]). Based on evidence and according to ACMG guidelines, this nonsense variant has been classified as pathogenic.

### 2.3.7. OFT-00786

This female patient was diagnosed at 7 years of age with low visual acuity and a family history of optic atrophy.

The patient is heterozygous for the mutation c.1034G>A(p.Arg345Gln) (NC_000003.11: g.193355069), which affects the dynamin domain producing a shift from a positively charged amino acid (arginine) to an uncharged polar amino acid (glutamine) at codon 345. This non-conservative change can lead to destabilization of the protein structure resulting in decreased protein function. In fact, numerous articles have reported that the change is associated with optic atrophy 1 [24,25]. In addition, computational prediction tools unanimously predict a deleterious effect on the gene, and its frequency in gnomAD population databases is extremely low. Based on this evidence, the c.1034G>A variant has been classified as pathogenic by ACMG guidelines.

**Table 2.** Genetic results of the probands for the *OPA1* gene according to *OPA1* transcript variant 8 (RefSeq: NM_130837.3) representing the longest transcript.

| Family ID | Individual ID | Sex | Relationship | Mutation | Location | ACMG Criteria * | ACMG Result | Coding Impact | Zygosity | Segregation Analysis Performed | De Novo/ Inherited | Reported by |
|---|---|---|---|---|---|---|---|---|---|---|---|---|
| OFT-00133 | 2620605 | M | Proband | NM_130837.3:c.267G>A (p.Trp89Ter) | Exon 2 | PVS1, PM2, PP1 | P | Nonsense | Hetero | Yes | Maternal | (Le Roux, 2019) [17] |
| OFT-00536 | 2624905 | M | Proband | NM_130837.3:c.2873_2876del (p.Val958GlyfsTer3) | Intron 28 | PVS1, PP2, PP5 | P | Frameshift | Hetero | Yes | Unknown | (Delettre, 2000) [9] (Wang, 2019) [18] |
| | 2235284 | M | Brother | | | | | | | | | |
| OFT-00560 | 3338850 | F | Proband | NM_130837.3:c.1406_1407del (p.Thr469LysfsTer16) | Exon 15 | PVS1, PM2, PP1 | P | Frameshift | Hetero | Yes | Paternal | Novel ** |
| | 3338857 | F | Brother | | | | | | Hetero | | | |
| OFT-00615 | 3370622 | M | Proband | NM_130837.3:c.1041_1043del (p.Val349del) | Exon 11 | PM1, PM2, PM4, PP1 | LP | In-frame | Hetero | Yes | Maternal | (Gallus, 2012) [19] (Kamakari, 2014) [20] |
| | 3395589 | M | Brother | | | | | | Hetero | | | |
| OFT-00641 | 3346308 | M | Proband | NM_130837.3:c.2013G>A (p.Trp671Ter) | Exon 22 | PVS1, PM2, PP1 | P | Nonsense | Hetero | Yes | Paternal | (Ferré, 2009) [21] |
| | 3366708 | M | Brother | | | | | | Hetero | | Paternal | |
| OFT-00677 | 3408438 | F | Proband | NM_130837.3:c.1099C>T (p.Arg367Ter) | Exon 11 | PVS1, PP5, PM2 | P | Nonsense | Hetero | No | Unknown | (Cardaioli, 2006) [22] (Chen, 2014) [23] (Pretegiani, 2017) [24] |
| OFT-00786 | 2854654 | F | Proband | NM_130837.3:c.1034G>A (p.Arg345Gln) | Exon 10 | PP5, PS3, PM2, PP3, PM1, PM5, PP2 | P | Missense | Hetero | No | Unknown | (Pretegiani, 2017) [24] (Liskova, 2016) [25] |

Abbreviations: P, pathogenic; LP, likely pathogenic; Hetero, heterozygous; Homo, homozygous; AD, autosomal dominant; ACMG, American College of Medical Genetics and Genomics. * ACMG Criteria in Appendix A, Table A1/ ** Not previously reported in the literature.

### 3. Discussion

DOA is classified as a rare disease (Orpha98672) given that its prevalence ranges from 1/12,000 to 1/50,000 [1]. However, DOA is the most commonly diagnosed inherited optic neuropathy and is usually diagnosed over the age of 16 years [3,26,27]. Our study focused on the pediatric population to determine whether they are more severely affected by DOA. Our results showed the lowest VA scores in children younger than 11 years. Above that age, the tendency was towards higher VA levels, similar to the adult population, in which a moderate VA decrease is usually reported [28]. The visual field of the patients who could perform the test reliably showed that cecocentral scotoma is a common finding among the patients [1,11], except for patient OFT-00677 who showed only a mild reduction in VA and a normal visual field.

Retinal ganglion cell (RGC) degeneration in patients with DOA preferentially involves the papillomacular bundle. There are two theories explaining this papillomacular bundle vulnerability, which contains the smallest RGCs in the retina, although the authors state these are insufficient for explaining the bundle's vulnerability to the OPA mutation. The first theory refers to the difference in the mitochondrial energy reserve within the small-caliber, parvocellular RGCs in the papillomacular bundle compared with the large-caliber, magnocellular RGCs. The second theory suggests that smaller RGCs produce more superoxides than detoxification [29].

OCT is a fast non-invasive imaging test that uses light waves to obtain quantitative and qualitative information of the retina and optic nerve. RNFL thickness appears to be reduced in patients with OPA1. Pretegiani et al. studied 60 Italian *OPA1* mutation carriers (52 symptomatic), belonging to 13 families, with a mean age of 37.6 years at the time of observation. Twelve cases were examined with OCT. The mean RNFL thickness was significantly reduced compared with that of the healthy controls, with a mean thickness of 69.5 ± 15 microns [24]. Asanad et al. compared the thinning pattern of retinal layers measured with OCT between patients with LHON and those with DOA [8] and showed a significant reduction of 69.7 ± 2.0 in RNFL thickness in the DOA subgroup similar to the chronic LHON cohort with preferential involvement of the temporal and inferior quadrants [30,31]. In our cohort, however, the mean RNFL thickness was 81.6 and 80.5 μm in the right and left eye, respectively. Such a reduction in thickness was not observed in our population, which could be explained by the already reported gradual reduction in RNFL thickness with age, given that these studies included adult patients [32]. In our study, the mean and temporal RNFL thickness were also thinner, but the difference was not statistically significant in the mean RNFL thickness in the group of children younger than 11 years.

OCT segmentation allowed us to obtain specific GCL measurements. Corajevic et al. [33] analyzed layer and location-specific retinal thickness deficits in patients with DOA and found that the reduction in retinal ganglion cell numbers in the nasal perifoveal area of the GCL combined with the attenuation of the temporal peripapillary RNFL area is the strongest diagnostic OCT marker of this condition. The mean baseline GCL was 52.5 and 52.4 μm in the right and left eye, respectively.

The first pathogenic mutations in *OPA1* were described in patients with ADOA by Delettre [9] and Alexander [34] in 2000. Thanks to advances in next-generation sequencing, 613 unique *OPA1* variants have since been reported to the *OPA1* locus specific database (NM_130837.2) (https://databases.lovd.nl/shared/genes/OPA1 (accessed on 20 October 2022)), 71% (436/613) of which have been classified as pathogenic, causing ADOA or related diseases. These mutations are distributed along the entire protein-coding sequence, but many are clustered in the GTPase domain. Most of these mutations are truncation variants, including the nonsense, frameshift, and splicing site variants, while approximately 40% (172/436) of them are missense variants, located predominantly in the GTPase domain. The 436 pathogenic mutations have been published in PubMed and are mostly associated with optic atrophy.

This study found seven mutations in the *OPA1* gene in our cohort. A novel pathogenic frameshift mutation (NM_130837.3: c.1406_1407del (p.Thr469LysfsTer16)) has been described for ADOA patients. Most of the mutations found are null variants and most of them are located in the GTPase domain (Figure 2).

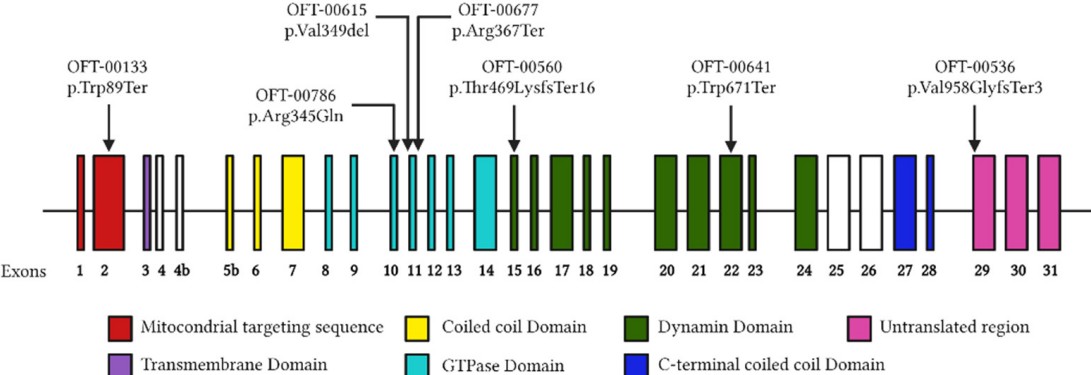

**Figure 2.** Distribution of *OPA1* variants. Each distinct variant observed in our cohort is represented above the respective exon. The structural domains of the protein encoded by each exon are detailed with different colors, respectively.

Based on the known statistics of reported *OPA1* variants, most of the variants found are truncation variants, as expected, encoding truncated forms of the protein; only one is a missense variant. In general, transcripts with premature termination codons are prone to degradation through sense-mediated mRNA decay [15], even constituting less than the expected 50% of the total set of *OPA1* transcripts, which could lead to haploinsufficiency, the main pathological mechanism implicated in OPA1-mediated disease [35]. Due to the energetic importance of OPA1 function in mitochondria, various studies have indicated that a significant loss of this function often indirectly sensitizes cells to mitochondrial apoptosis and other forms of cell death by compromising the bioenergetic integrity of the mitochondrion, ATP levels, and its structure [36,37]. This evidence supports the causality of the variants found in *OPA1* in our patients with ADOA.

In reference to the location in the structure of the *OPA1* gene, the p.Trp89Ter mutation is located in the mitochondrial targeting sequence, which could prevent the targeting of the protein to the mitochondrial membrane, prevent further processing to L-OPA1 and S-OPA1 and, consequently, influence the functionality of the protein. Grau, Tanja et al. [38] demonstrated by functional analysis for optic atrophy 3 (*OPA3)* that mutations were located in the MTS results in altered steady-state levels and fragmented mitochondrial network. This, coupled with possible degradation to sense-mediated mRNA decay, could support our hypothesis of variant causation. None of the remaining variants affect the S1 or S2 sites, so we are not able to analyze the established relationship between the imbalance of the L-OPA1 and S-OPA1 forms and optic atrophy.

In addition, three of the variants found (p.Val349del, p.Arg367Ter and p.Arg345Gln) (42%) affect the GTPase domain, which is responsible for the main function of the dynamin: the fusion of mammalian inner mitochondrial membranes (Figure 2). The GTPase domain is a highly conserved domain and impairment of its activity has been shown to cause instability of the inner mitochondrial membrane structure and unbalanced proton leakage leading to decreased membrane potential [5,39]. Therefore, it would be expected that mutations located in the GTPase domain of our patients are responsible for the phenotype. However, most of the variants located in the GTPase domain described in the literature have been associated with ADOA plus, a more complex neurodegenerative disorder with extraocular manifestations [40,41]. This duality in the findings in the literature and our study, coupled with the incomplete penetrance described for *OPA1* and the high phenotypic interindividual variability observed in patients, makes unequivocal genotype–phenotype correlation difficult [10,42,43].

## 4. Materials and Methods

We conducted a retrospective review of the medical records of 11 children with confirmed *OPA1* DOA treated in the Department of Pediatric Ophthalmology in La Paz University Hospital between 1 January 2018 and 1 January 2022. Informed consent was obtained from all of the patients' parents or guardians, and samples from the patients and probands were acquired in accordance with the principles of the Declaration of Helsinki.

The main inclusion criteria were a diagnosis of DOA with a confirmed mutation in *OPA1* and an age at diagnosis younger than 16 years. Patients with a phenotype associated with DOA plus were excluded.

The demographic data collected from the 11 patients during this study were sex, race, medication compliance, age at diagnosis, and family history reported by the patients' parents/guardians.

### 4.1. Genetic Studies

For the genetic analysis, genomic DNA was isolated from leukocytes in peripheral venous blood samples in our institute's preanalytical area using the commercial Chemagic Magnetic Separation Module I (Chemagen, PerkinElmer, Waltham, MA, USA). DNA concentrations were measured by spectrofluorometer quantification using a NanoDrop ND-1000 Spectrophotometer (ThermoFisher Scientific, Waltham, MA, USA). Paired-end libraries were created using 1 μg of genomic DNA with KAPA HyperPrep Kit (Roche, NimbleGen, Inc., Pleasanton, CA, USA) and hybridization with a KAPA HyperCapture Reagent Kit (Roche, NimbleGen, Inc., Pleasanton, CA, USA).

Capturing was performed using a custom panel (OftSd-v3.1) (Appendix A), and sequencing was performed on the Illumina HiSeq 4000 platform (Illumina, Inc., San Diego, CA, USA). The data produced after sequencing were aligned and mapped to the human genome reference sequence (GRCh37/hg19).

The OftSd-v3.1 panel was designed with NimbleDesign software (Roche NimbleGen, Inc., Pleasanton, CA, USA): HG19 NCBI Build 37.1/GRCh37. The target bases covered 98.63% and the size was 2,888,420 Kb.

The first analysis was a bioinformatic analysis performed by the clinical bioinformatics team of the Institute of Medical and Molecular Genetics. The bioinformatic analysis was conducted using an analytical algorithm designed by the bioinformatics team, enabling the identification of point polymorphisms (single nucleotide polymorphisms) and insertions and deletions of small DNA fragments (indels) in the regions that were captured by the NGS panels.

The system comprises a sample pre-processing phase, alignment of reads with a reference genome, identification and functional annotation of variants, and variant filtering. All phases employ proprietary tools and open tools widely used in the scientific community. All phases are robustly designed and include statistical parameters that provide information on the process status and assess the desirability of further analysis. The monitoring of the process and the quality controls performed by the system allow a reliable report to be issued on previous variations.

The bioinformatics analysis was performed using the following software tools: trimmomatic-0.36, bowtie2-align version 2.0.6, picard-tools 1.141, samtools version 1.3.1, bedtools v2.26 and GenomeAnalysisTK version 3.3-0. The databases employed were db-NSFP version 3.5, dbSNPv151, ClinVar date 20180930, ExAC-1, SIFT ensembl 66, Polyphen-2 v2.2.2, MutationAssessor, release 3, FATHMM, v2.3, CADD, v1.4, and dbscSNV1.1.

Below genotype–phenotype correlation was carried out. The pathogenic clinical significance of variants found in the patients was evaluated by employing the following databases: (1) Varsome and Franklin Genoox, which contain an algorithm using the 2015 ACMG guidelines [44], to classify the variants as pathogenic, likely pathogenic, variants of uncertain significance, likely benign, or benign; (2) PubMed and the University of California Santa Cruz Genome Browser Home, on which a thorough literature search was performed;

and (3) the Human Gene Mutation Database Professional, GnomAD, LOVD, ClinVar and PubMed to check whether the variants had been previously reported.

### 4.2. Ophthalmological Studies

A complete ophthalmological evaluation was performed on all patients during each visit. Baseline and final VA were assessed using the decimal scale. Low vision was defined as a VA $\leq$ 0.3. Colour vision was assessed with the Farnsworth-Munsell 100 Hue test, and intraocular pressure was measured with the iCare TA01i tonometer. If there were associated systemic symptoms, these were included in the evaluation and the patients classified as DOA plus.

Cirrus HD-OCT (Carl Zeiss Meditec, Dublin, CA, USA) imaging was obtained from all patients, comparing the first and last visit reliable images. Mean RNFL and temporal sector thickness were examined, as well as the mean thickness of the ganglion cell layer. We also measured the central macular thickness. These data were compared with our own pediatric patient database. Visual field testing was performed on the cooperative patients using the 24-2 Humphrey test. At least two visual fields with the same defect were needed to confirm a defect. Optomap readings were obtained for 11 patients with the Optos Daytona Digital Retinal Scanner.

For the statistical analysis, the right eye of each patient was randomly selected. RNFL and GCL thicknesses were compared between the control participants and the patients with DOA using the chi-squared non-parametric test.

### 5. Conclusions

In conclusion, the pathogenic mutations in *OPA1* causing DOA were mainly truncating variants and are generally found in the GTPase domain following the trend presented to date in previous studies. Furthermore, the bioinformatics tools employed, along with the clinical evaluation of the variants, have enabled us to describe a new variant in *OPA1:* NM_130837.3: c.1406_1407del (p.Thr469LysfsTer16) responsible for our patients' phenotype, showing the importance of genetic studies as a diagnostic method.

The phenotypic variability presented in our cohort challenges the establishment of a clear phenotype–genotype relationship for the *OPA1* gene. Patients will need longer follow-ups to confirm the phenotype severity discrepancy found in our cohort. Further studies are needed to corroborate this correlation.

It would therefore be interesting to expand the number of patients analyzed to establish a clearer genotype–phenotype correlation between them and try to carry out functional studies to study the impact of the novel mutation of *OPA1* in the eye's development and function.

**Author Contributions:** Conceptualization, N.A., P.R.-S., E.V. and S.N.; formal analysis, Á.d.P.; investigation, N.A. and P.R.-S.; methodology, M.N.-M. and V.E.F.M.; resources, F.S.-S., E.R.-O. and L.D.-M.; supervision, E.V. and S.N.; validation, N.A., P.R.-S and M.G.-C.; visualization, N.A., P.R.-S., E.V. and S.N.; writing—original draft, N.A. and P.R.-S.; writing—review and editing, E.V. and S.N. All authors have read and agreed to the published version of the manuscript.

**Funding:** Project "18/1234", funded by Instituto de Salud Carlos III (ISCIII) and co-funded by the European Union and by a grant from ONCE (grant number 2020/0197782).

**Institutional Review Board Statement:** The study was conducted according to the guidelines of the Declaration of Helsinki and approved by the Ethics Committee of La Paz University Hospital of Madrid (protocol code PI-4016 approved on the 3 February 2020).

**Informed Consent Statement:** Informed consent was obtained from all participants involved in the study.

**Acknowledgments:** We are grateful to the patients and their families.

**Conflicts of Interest:** The authors declare no conflict of interest.

## Appendix A

**Table A1.** American College of Medical Genetics and Genomics criteria.

| Code | Description |
|------|-------------|
| PVS1 | Null variant (nonsense, frameshift, canonical $\pm 1$ or 2 splice sites, initiation codon, single or multiexon deletion) in a gene where loss of function is a known mechanism of disease. |
| PS1 | Same amino acid change as a previously established pathogenic variant regardless of nucleotide change. |
| PS2 | De novo (both maternity and paternity confirmed) in a patient with the disease and no family history. |
| PS3 | Well-established in vitro or in vivo functional studies supporting a damaging effect on the gene or gene product. |
| BS1 | Allele frequency is greater than expected for the disorder. |
| PM1 | Located in a mutational hotspot and/or critical and well-established functional domain (e.g., active site of an enzyme) with no benign variation. |
| PM2 | Absent from controls (or at extremely low frequency if recessive) in Exome Sequencing Project, 1000 Genomes Project, and Exome Aggregation Consortium. |
| PM3 | For recessive disorders, detected in trans with a pathogenic variant. |
| PM4 | Protein length changes as a result of in-frame deletions/insertions in a non-repeat region or stop-loss variants. |
| PM5 | Novel missense change at an amino acid residue where a different missense change determined to be pathogenic has been previously observed. |
| PP1 | Cosegregation with disease in multiple affected family members in a gene definitively known to cause the disease. |
| PP2 | Missense variant in a gene that has a low rate of benign missense variation and in which missense variants are a common mechanism of disease. |
| PP3 | Multiple lines of computational evidence support a deleterious effect on the gene or gene product (conservation, evolutionary, splicing impact, etc.) |
| PP5 | Reputable source recently reports variant as pathogenic, but the evidence is not available to the laboratory to perform an independent evaluation. |
| BP4 | Multiple lines of computational evidence suggest no impact on gene or gene product (conservation, evolutionary, splicing impact, etc.) |
| BP6 | Reputable source recently reports variant as benign, but the evidence is not available to the laboratory to perform an independent evaluation. |

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
