# Peer review of "OPA1 Dominant Optic Atrophy: Diagnostic Approach in the Pediatric Population"

_cimb, doi:10.3390/cimb45010030_

Round 1

Reviewer 1 Report

Currently it is confirmed that 60-80% of ADOA cases are caused by OPA1 gene mutations. It is worth to mention at least 20 nuclear genes involved in etiology and to describe genetic heterogeneity of ADOA. Especially ACO2 and WFS1 genes are responsible for approx 12% of cases each.

It is very surprising that in all families the OPA1 gene mutation was found. It is  not known if the families tested were the only families with optic atrophies or they were choosen on any way.

The group is very small so no final coclusions can be done. These are just the preliminary results.

In my opinion, for several reasons, all the data regarding the off-label treatment with idebenone should be deleted from this paper:

- the treated group was extremely small (6 patients),

- there was no control group,

- stabilization of the visual acuity is a normal phenomenon in many patients,

- idebenone is a benzoquinone which allows to omit e defective complex I and transfer electrons directly to complex III of the mitochondrial electron transport chain, and there are no sceintific data supporting its usage in ADOA (OPA1 codes for a GTPase...)

Round 2

Reviewer 1 Report

I still think that there is no justification to describe the off-label treatment with idebenone because of the reasons metioned before. There are no explanations of mechanism of action in ADOA and no convincing results of earlier trials. This part must be deleted.

Author Response

REVIEWS

Manuscript ID: cimb-2081676

Thank you very much for reviewing the manuscript. The changes made are marked in the paper attached.

Review 1

Comments and Suggestions for Authors

I still think that there is no justification to describe the off-label treatment with idebenone because of the reasons metioned before. There are no explanations of mechanism of action in ADOA and no convincing results of earlier trials. This part must be deleted.

Reply: Thank you for your appreciation and your point of view. Finally, we have decided to remove idebenone related information as you requested. Therefore, we have modified several parts of the text indicated in the manuscript, including the title, to fit the theme of the paper. It is true that the idebenone data could be useful over a longer period of time when the cohort of treated patients is larger and allows us to obtain meaningful treatment results.

Thank you very much for the comments and the review, it has helped us to improve the article and the research.